

# Cities may save some threatened species but not their ecological functions

Álvaro Luna, Pedro Romero-Vidal, Fernando Hiraldo and Jose L. Tella

Department of Conservation Biology, Estación Biológica de Doñana, CSIC, Sevilla, Spain

Corresponding authors
Álvaro Luna, alvalufer@gmail.com
Jose L. Tella, tella@ebd.csic.es

## ABSTRACT

**Background:** Urbanization is one of the main causes of biodiversity loss worldwide. Wildlife responses to urbanization, however, are greatly variable and, paradoxically, some threatened species may achieve much larger populations in urban than in natural habitats. Urban conservation hotspots may therefore help some species avoid regional or even global extinctions, but not conserve their often overlooked ecological functions in the wild. We aim to draw attention to this issue using two species of globally threatened parrots occurring in the Dominican Republic: the Hispaniolan amazon (*Amazona ventralis*) and the Hispaniolan parakeet (*Psittacara chloropterus*).

**Methods:** We conducted a large-scale roadside survey in June 2017 across the country to estimate the relative abundance of parrots in natural habitats, rural habitats, and cities. We combined this with informal interviews with local people to collect information on past and current human impacts on parrot populations. We also looked for foraging parrots to assess their potential role as seed dispersers, an ecological function that has been overlooked until very recently.

**Results:** Relative abundances of both parrot species were negligible in rural areas and very low in natural habitats. They were generally between one and two orders of magnitude lower than that of congeneric species inhabiting other Neotropical ecosystems. Relative abundances were six times higher in cities than in natural habitats in the case of the Hispaniolan parakeet and three times higher in the case of the Hispaniolan amazon. People indicated hunting for a source food and to mitigate crop damage as causes of parrot population declines, and a vigorous illegal trade for parrots (131 individuals recorded, 75% of them poached very recently), mostly obtained from protected areas where the last small wild populations remain. We observed parrots foraging on 19 plant species from 11 families, dispersing the fruits of 14 species by carrying them in their beaks and consuming them in distant perching trees. They discarded undamaged mature seeds, with the potential to germinate, in 99.5% of cases ($n = 306$), and minimum dispersal distances ranged from 8 to 155 m (median = 37 m).

**Discussion:** The loss of ecological functions provided by some species when they disappear from natural habitats and only persist in cities may have long-term, unexpected effects on ecosystems. Our example demonstrates how two cities may soon be the last refuges for two endemic parrots if overharvesting continues, in which case their overlooked role as seed dispersers would be completely lost in nature. The functional extinction of these species could strongly affect vegetation communities in an island environment where seed-dispersal species are naturally

scarce. While conservation plans must include urban populations of threatened species, greater efforts are needed to restore their populations in natural habitats to conserve ecological functions.

## INTRODUCTION

Urbanization is one of the most rapidly growing, prevalent and lasting causes of habitat change (*Seto, Güneralp & Hutyra, 2012*) and results in the loss of biodiversity through local extinction processes worldwide (*McKinney, 2006*; *Grimm et al., 2008*; *Sol et al., 2014*). However, wildlife responses to urbanization are highly variable: while most species are unable to occupy these new habitats, including endemic and threatened species (*González-Oreja, 2010*), others are able to persist or colonize even the most populated cities (*Chace & Walsh, 2006*; *Kark et al., 2007*; *Carrete & Tella, 2011*; *Sol et al., 2014*). Paradoxically, some species achieve much higher densities in urban than in natural habitats and thus, cities arise as conservation hotspots (*Mason, 2010*; *Rebolo-Ifrán, Tella & Carrete, 2017*). Cities have also been known to host some threatened species, thus becoming key refuges to guarantee their persistence. As recent examples, 22% of the known occurrences of threatened plants in the USA fall within the 40 largest cities (*Schwartz, Jurjavcic & Joshua, 2002*), a third of 54 cities sampled by *Aronson et al. (2014)* contained globally threatened birds, and Australian cities support more threatened plant and animal species than all other non-urban areas on a unit-area basis (*Ives et al., 2016*). Thus, the conservation of some threatened species is an additional motivation to make urban environments compatible with the persistence of wildlife (*Miller & Hobbs, 2002*; *Dearborn & Kark, 2010*).

From a species-based conservation point of view, the fact that cities may help some species avoid regional or even global extinctions (*Ives et al., 2016*; *Gibson & Yong, 2017*) is a welcomed insight for conservationists and policymakers. However, little attention has been paid to the ecological consequences of the restriction of species to urban environments as their final refuges. Particularly, the loss of ecological functions and ecosystem services provided by species when they disappear from natural habitats to only persist in urban environments may have long-term, unexpected effects on ecosystems. We aim to draw attention to this likely phenomenon using two species of threatened parrots as an example.

Parrots (Psittaciformes) are among the most threatened bird orders with 28% of the c. 400 species of the world classified as threatened by the IUCN Red List (*Olah et al., 2016*). Habitat loss, through logging and the spread of agriculture, is one of the main threats to parrots worldwide (*Olah et al., 2016*). However, some parrot traits (such as their high inter-individual variability in behavior associated with their relatively large brain sizes, *Carrete & Tella, 2011*) allow them to successfully colonize and thrive in urban

habitats, where they make use of the new food and nesting resources provided by cities (*Fitzsimons et al., 2003*; *Davis, Taylor & Major, 2011*; *Tella et al., 2014*). On the other hand, parrots are highly valued as pets given their attractive coloration and ability to imitate human speech. This societal demand drives an intense domestic and international trade that contributes to the rarity in the wild of many parrot species (*Tella & Hiraldo, 2014*; *Annorbah, Collar & Marsden, 2016*; *Berkunsky et al., 2017*), but also to the establishment of urban parrot populations both within their native and foreign distribution ranges due to the escape or deliberated release of caged parrots (*Abellán et al., 2017*; *Cardador et al., 2017*; *Mori et al., 2017*). This has created a dilemma: while many wild parrot populations are vanishing due to habitat loss and harvesting for the pet trade, some exotic urban populations are flourishing worldwide to the point that their conservation value could overcome concerns about their potential negative impacts as invasive species (*Gibson & Yong, 2017*). Conservation efforts focused on some urban parrot populations may save those species from extinction (*Gibson & Yong, 2017*), but not their ecological functions in their native ecosystems, which remain poorly understood. Parrots have been traditionally considered as pure plant antagonists, as they consume fruits, seeds, flowers and other non-reproductive tissues of their food plants. Recent works, however, have suggested that several parrot species also act as plant mutualists through pollination, seed dispersal, and plant healing (*Blanco, Hiraldo & Tella, 2018*). While the role of parrots as seed dispersers has been largely overlooked (*Tella et al., 2015*; *Blanco et al., 2016*), long-distance seed dispersal by parrots has been shown to be key in some ecosystems (*Blanco et al., 2015*; *Tella et al., 2016a*, *2016b*; *Baños-Villalba et al., 2017*). Parrots may even shape vegetal landscapes by influencing the spatial distribution and demography of their food plants and vegetal communities (*Blanco et al., 2015*; *Baños-Villalba et al., 2017*; *Speziale et al., 2018*), and plant-parrot mutualistic interactions may lead to an increase in the robustness of parrot communities when facing the loss of plant species (*Montesinos-Navarro et al., 2017*). All of these ecological functions may be lost when human persecution causes the disappearance of parrots in the wild, with remaining population confined to urban refuges.

Here, we used as a model the two threatened Caribbean parrot species endemic to Hispaniola Island: the Hispaniolan parakeet (*Psittacara chloropterus*) and the Hispaniolan amazon (*Amazona ventralis*) (Fig. 1). These species are listed by the IUCN Red List as Vulnerable due to their population declines associated with habitat loss, hunting and poaching for the pet trade (*BirdLife International, 2016a*, *2016b*). In fact, these species are known to be trapped and hunted since the Spanish colonization of the island (*Wiley & Kirwan, 2013*). We therefore hypothesized that (1) past and current human persecution of parrots may cause a contraction of wild populations to predator-free (considering humans as predators through hunting and trapping for the pet trade) urban refuges (*Rebolo-Ifrán, Tella & Carrete, 2017*), and (2) that these parrot species may act as legitimate seed dispersers of several plant species (*Blanco, Hiraldo & Tella, 2018*), an ecological function thus far overlooked that might be definitively lost if these species continue to disappear in the wild. For testing these hypotheses, we conducted a large-scale field survey to estimate parrot abundances in natural, rural and urban habitats, obtained

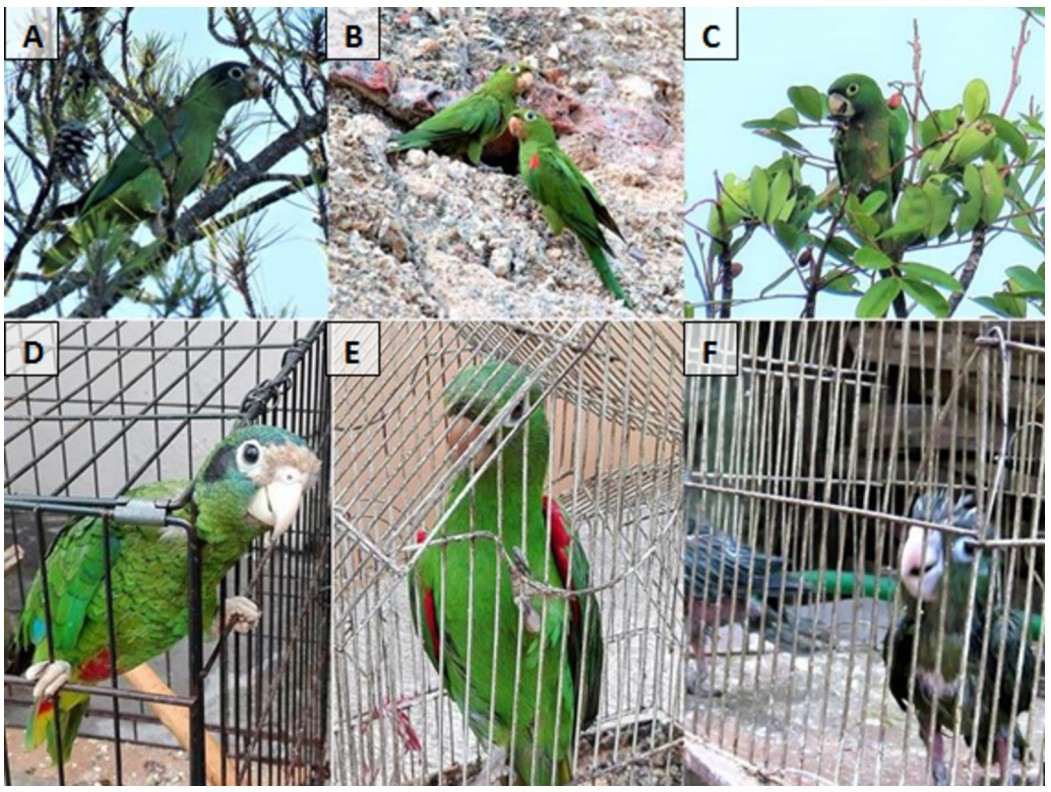

**Figure 1 Parrot species.** Three species of parrots occur in the Dominican Republic: (A) the Hispaniolan amazon (*A. ventralis*), (B) the Hispaniolan parakeet (*P. chloropterus*), and (C) the Jamaican parakeet (*Eupsittula nana*). The three species are illegally trapped for sale as pets (D, E and F, respectively). Photo credits: Á. Luna (A, C, F) and J.L. Tella (B, D, E).

information from local people to know whether hunting and trapping are still threating these species, and assessed their potential role as seed dispersers.

## MATERIALS AND METHODS

### Study area and species

The Dominican Republic is located in the central area of the Caribbean Sea, covering two thirds of La Hispaniola Island (the rest corresponding to Haiti). Since Spanish colonization in 1492, the island habitats have transformed to farmland, with agriculture currently covering 48.7% of its surface area (*World Bank, 2014*), although habitat degradation was higher in past decades and centuries (*Olivo, 2007*). Despite the habitat loss and fragmentation, this country still holds the largest variety of ecosystems in the Caribbean region, due to its mountainous relief (altitude ranging from 0 to 3,098 m.a.s.l.), climate variability, and a large surface area (25,472 km$^2$) covered by 119 protected areas (c. 50% of the country). The main ecosystems can be broadly grouped as tropical rainforests in the northern lowlands, coniferous forests in the central mountains, and tropical dry forests in the southern lowlands. By 2001, 65% of the population (8.7 million people) lived in urban areas, with the capital (Santo Domingo) holding 3.5 million people

and an urban population growth rate that reached 2.3% between 2000 and 2005 (http://www.nationsencyclopedia.com/Americas/Dominican-Republic-POPULATION.html). The Dominican Republic constitutes the core area for the two endemic Hispaniolan parrot species (*P. chlopterus* and *A. ventralis*), which are considered near extinction in Haiti (*BirdLife International, 2016a*, *2016b*). A third, Near Threatened species endemic to Jamaica, the Jamaican parakeet (*Eupsittula nana*) (*BirdLife International, 2016c*), has been reported in the Dominican Republic, but its status as a native or alternatively human-mediated introduced species remains uncertain (*Latta et al., 2006*).

## Estimating habitat-related parrot abundances

Habitat use by parrots was assessed through large-scale roadside surveys, conducted under permits from the Ministry of the Environment of the Dominican Republic. Among different methods available to estimate parrot abundance, roadside surveys are recommended given that they allow to sample large areas and thus increase the likelihood of recording parrots, as they usually show low densities and aggregated distributions resulting from their highly-mobile and flocking behavior (*Dénes, Tella & Beissinger, 2018*). This methodology has previously allowed the assessment of parrot abundances (*Grilli et al., 2012*; *Blanco et al., 2015*; *Tella et al., 2016a*; *Baños-Villalba et al., 2017*), of their foraging behavior (*Tella et al., 2016b*; *Montesinos-Navarro et al., 2017*), and of the effects of habitat loss and fragmentation in a variety of raptor and parrot species in different Neotropical biomes (*Carrete et al., 2009*; *Tella et al., 2013*). One of the authors (AL) travelled throughout the country in June 2016, gathering information on the distribution of habitats across the island. Thereafter, using recent satellite maps, we selected a network of secondary, mostly unpaved roads to be surveyed, which covered natural habitats as well as agricultural land and urbanized areas throughout the country. We included roads that crossed 12 protected areas (including the National Parks del Este, Jaragua, Bahoruco, Sierra de Neiva and Cordillera Central), where natural ecosystems of the island are still well preserved. The road network was plotted on paper maps and uploaded on a portable GPS. We conducted the field survey from June 6 to June 21, 2017, at the end of the parrots' breeding season (*Latta et al., 2006*), with 2,143.5 km surveyed (Fig. 2). Surveys were conducted in the morning and afternoon by three to four persons driving a car at low speed (10–25 km/h). Habitats crossed through the roadside transects were grouped into the main island habitats: (1) coniferous forests, (2) tropical rainforests, (3) tropical dry forests, (4) farmland, (5) small villages, and (6) large cities. We recorded every time the habitat changed from one type to another to measure the length (in km) of the habitat crossed by the transect (sub-transect). For each sub-transect we also recorded the number of observed parrots of each species. We briefly stopped the car every time parrots were detected to record the species, number of individuals, their activity, the distance at which they were detected (detection distance) using a laser rangefinder (Leica Geovid 10 × 42, range: 10–1,300 m), and to geo-reference the site. All roadside transects were surveyed only once.

The estimation of parrot densities through distance sampling modelling (*Blanco et al., 2015*; *Baños-Villalba et al., 2017*) was not possible due to the scarcity of the species

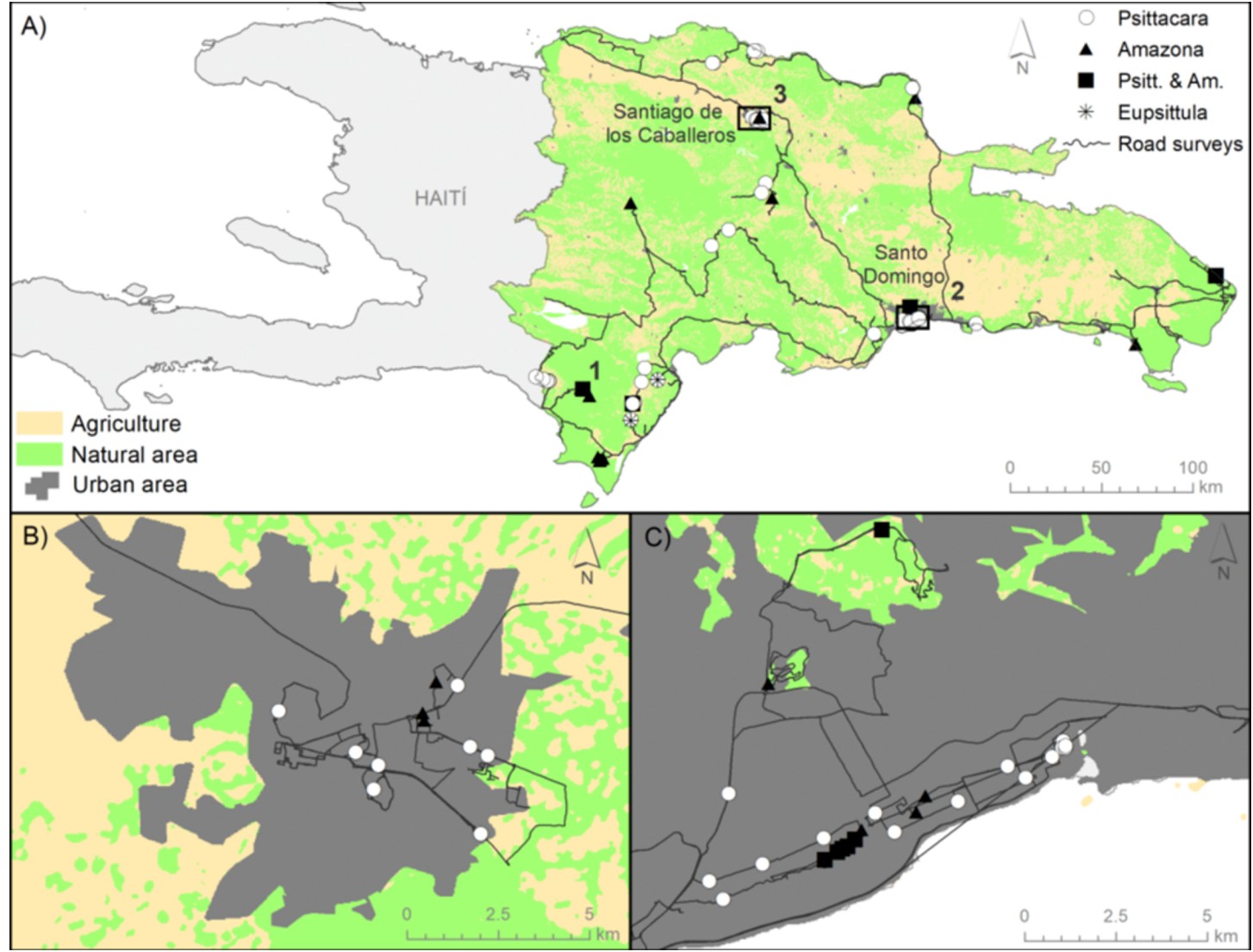

**Figure 2 Location of study area, surveys, and parrot records.** (A) Map of the Hispaniolan island including the roadside transects surveyed in Dominican Republic (black lines, totaling 2134.5 km), the records of *P. chloropterus* (white dots) and *A. ventralis* (black triangles), both species together (black squares), and *E. nana* (asterisks). Numbers 1, 2 and 3 show the location of communal roosts. (B) and (C) show details for the main cities, Santiago and Santo Domingo, respectively. The main habitats depicted (agriculture, natural and urban areas) were obtained from the Caribbean Land Cover Project (https://lca.usgs.gov/carland/index.php).               

and thus the very low number of contacts we obtained for both species of parrots in most habitats (see Results), then precluding obtaining reliable habitat-specific detectability functions (see *Dénes, Tella & Beissinger, 2018*, for distance sampling modelling requirements). The number of contacts and associated detection distances of parrots were also low for distance sampling modeling when pooling habitats into three main categories: (1) large cities, (2) rural habitats (grouping villages and farmland, i.e., a mosaic of small villages and houses embedded in agricultural lands), and (3) natural habitats (grouping coniferous, tropical rainforest and tropical dry forests). We thus relied on an index of relative abundance of each species in each habitat as the number of individuals

recorded/number of km surveyed (*Tella et al., 2013*, *2016b*; *Blanco et al., 2015*). Nonetheless, results obtained through this index were highly correlated with those obtained through distance sampling modelling in other parrot communities (*Dénes, Tella & Beissinger, 2018*). Differences in relative abundances of each parrot species among large cities, rural and natural habitats were tested using generalized linear models (GLM, Poisson error distribution and log link function), with the number of individuals recorded in each sub-transect as the response variable, the length (in kilo meter) of each sub-transect as a covariate, and habitat as a fixed factor (*Tella et al., 2013*). The obtained results could be biased due to potential biases in distances of detection, which ranged between 0 and 1,200 m ($n = 76$). Parrots seemed to do not avoid the proximity of the roads surveyed. In fact, 15% of the observations were recorded just in the border of the roads (distances close to 0 m), and 35% of the observations were recorded at distances $\leq 10$ m from the roads. On the other hand, a GLM (with log-transformed detection distances as response variable, normal distribution and identity function) showed no differences between species (Wald $\chi^2 = 0.91$, d$f = 1$, $p = 0.327$) nor an effect of flock size (Wald $\chi^2 = 1.48$, d$f = 1$, $p = 0.233$), with a marginally significant effect of habitat (Wald $\chi^2 = 5.11$, d$f = 2$, $p = 0.078$). This effect resulted from a slightly higher detectability of parrots in rural (median detection distance = 68.5 m, range = 5–132 m) and natural habitats (median = 80.0 m, range = 15–1,200 m) than in large cities (median = 66.0, range = 0–350 m). From our experience, the highly intense and noisy car traffic and presence of tall buildings made difficult both the oral and visual detection of parrots in cities at larger distances. If anything, this slight bias in detectability would underestimate the relative abundance of parrots in cities compared to natural habitats, thus making our results conservative.

In addition to the above systematic survey, we also looked for flocks of parrots flying at sunset to potential communal roosts. When we located a communal roost, three to five people were situated at vantage points to count the number of individuals gathering at the roost site until night time. This information on roost sizes could help future long-term monitoring programs of these species (*Dénes, Tella & Beissinger, 2018*).

### Recording conservation threats

We wanted to know whether the main conservation threats highlighted in the literature, namely poaching for the pet trade and hunting (*Latta et al., 2006*; *BirdLife International, 2016a*, *2016b*), still affect parrot populations in the Dominican Republic. We discarded the possibility of conducting a systematic survey using questionnaires (see methodological recommendations by *Young et al., 2018*), given that most people could be reluctant to respond as hunting, trapping and trading parrots are illegal activities in Dominican Republic since 2000. Therefore, to answer our simple question (i.e., whether these threats are still alive in the country despite of prohibitions), we simply stopped in different villages while travelling across the country to informally converse with local people we found in the streets. We presented ourselves as foreign ornithologists interested in the observation and conservation of birds and especially of parrots, and thus people viewed us as bird watching tourists that could not constitute a threat for their potential illegal

activities. During these informal talks, we asked people about the current or past presence of parrots in the surroundings and their conservation problems. The fact that we speak the same language (Spanish) surely facilitated the long and friendly conversations we had, thus gaining confidence on the information obtained (*Young et al., 2018*). Most people freely responded, providing their perception of past and current status of parrot populations and their threats, and often guided us to the homes of neighbors, family or friends who kept parrots as pets. In such cases, we were allowed by owners to take pictures of their pets and often received voluntary much more information than we expected, including the age of pets, where and how they were obtained, and how much they paid for them. In some cases people also provided details on other threats such as hunting as a food source or to avoid crop damage. We did not record the names of informants, so their identities remain anonymous.

### Recording seed dispersal

During and outside of the roadside surveys, we looked for foraging parrots to record seed dispersal through stomatochory, i.e., when parrots fly from a fruiting plant carrying fruits with the beak to handle and consume them in a distant perching tree, then measuring dispersal distances with a laser rangefinder (*Tella et al., 2015*, *2016a*, *2016b*; *Blanco et al., 2015*; *Baños-Villalba et al., 2017*). We also looked below the identified perching trees for dispersed fruits and seeds that parrots dropped after consumption, and thus estimated a minimum dispersal distance for each seed as the distance to the nearest fruiting plant of the same species (see *Tella et al., 2016b*). All perching trees were species different to the plant species that produced the dispersed fruits, so there is not the possibility of making mistakes by recording seeds naturally falling to the ground or handled by parrots in the mother tree. We examined every dispersed fruit or seed to determine the proportion of mature, intact seeds (parrots often consumed the fruit pulp and discarded entire seeds) that had the potential to germinate after dispersal by parrots.

## RESULTS

### Habitat-related parrot abundances

Despite our large-scale roadside survey, covering 2,143.5 km, we only obtained 58 records of Hispaniolan parakeets and 18 records of Hispaniolan amazons throughout the country (Fig. 2), totaling 438 and 71 individuals, respectively (Table 1). Most of the individuals were recorded in cities, where the largest relative abundances were obtained (Table 1). Only Hispaniolan amazons reached relative abundances close to that found in large cities in two natural habitats (tropical rain and dry forests, see Table 1). Regarding the Jamaican parakeet, we only recorded this species through systematic roadside surveys in two localities of Sierra de Bahoruco (Fig. 2), totaling 13 individuals. The scarcity of this species precluded an examination of its habitat-related abundances.

The low number of contacts obtained through the systematic roadside surveys in habitats other than cities (Table 1) led us to pool habitats into three categories for statistical analyses: (1) large cities, (2) rural habitats (grouping villages and farmland, i.e., a mosaic of small villages and houses embedded in agricultural lands), and (3) natural

**Table 1 Raw results of the roadside survey.**

| Habitat | P. chloropterus | | | | A. ventralis | | |
|---|---|---|---|---|---|---|---|
| | km | Nindv | Nrec | Indv/km | Nindv | Nrec | Indv/km |
| City | 370.34 | 262 | 38 | 0.7 | 25 | 10 | 0.06 |
| Village | 509.38 | 17 | 4 | 0.03 | 4 | 1 | $7.85 \times 10^{-3}$ |
| Farmland | 375.39 | 12 | 1 | 0.03 | 1 | 1 | $2.66 \times 10^{-3}$ |
| Coniferous forest | 108.64 | 48 | 2 | 0.44 | 0 | 0 | 0 |
| Tropical rain forest | 314.46 | 70 | 10 | 0.22 | 19 | 3 | 0.06 |
| Tropical dry forest | 465.29 | 29 | 3 | 0.06 | 22 | 3 | 0.04 |
| TOTAL | 2143.5 | 438 | 58 | | 71 | 18 | |

Note:
Number of km surveyed in each habitat, number of individuals (Nindiv), number of records (Nrec) and relative abundance (number of individuals/km surveyed) obtained for the Hispaniolan parakeet (*P. chloropterus*) and Hispaniolan amazon (*A. ventralis*).

**Table 2 Relative abundances of parrots.**

| | Estimate | SE | 95% CI | Wald $\chi^2$ | P |
|---|---|---|---|---|---|
| **P. chloropterus** | | | | | |
| Intercept | −1.951 | 0.46 | [−2.85, −1.05] | 17.96 | 0.000 |
| City | 3.297 | 0.49 | [2.33, 4.27] | 44.50 | 0.000 |
| Natural | 1.542 | 0.50 | [0.57, 2.52] | 9.63 | 0.002 |
| Rural | 0 | – | – | – | – |
| Subtransect length | 0.039 | 0.01 | [0.03, 0.05] | 40.28 | 0.000 |
| **A. ventralis** | | | | | |
| Intercept | −3.625 | 0.53 | [−4.67, −2.58] | 45.93 | 0.000 |
| City | 3.008 | 0.59 | [1.85, 4.17] | 25.79 | 0.000 |
| Natural | 2.023 | 0.56 | [0.92, 3.12] | 12.97 | 0.000 |
| Rural | 0 | – | – | – | – |
| Subtransect length | 0.031 | 0.01 | [0.2, 0.05] | 17.78 | 0.000 |

Note:
Results of generalized linear models showing differences among habitats in the abundance of the two parrot species while controlling for the length (in kilo meter) of each subtransect.

habitats (grouping coniferous, tropical rainforest and tropical dry forests) (see Methods). The models obtained showed a strong effect of habitat on the abundance of both wild Hispaniolan parakeets and amazons while controlling for the length of each sub-transect surveyed (Table 2). The resulting estimated marginal means indicate a very low abundance of both species in rural habitats, while their abundance was six times higher in cities than in natural habitats in the case of the Hispaniolan parakeet and three times higher in the case of the Hispaniolan amazon (Table 3).

We censused the urban communal roost of Hispaniolan parakeets in Santo Domingo found by one of the authors (AL) in 2016, and recorded 1,580 individuals at sunset. We could not find the urban communal roost for this species in Santiago, but we found a roost of Hispaniolan amazons with about 50 individuals. We were informed of only one communal roost in natural habitats, located in Sierra de Bahoruco, one of the

**Table 3 Differences in relative parrot abundances among habitats.**

| | A. ventralis | | P. chloropterus | |
|---|---|---|---|---|
| | **Mean** | **95% CI** | **Mean** | **95% CI** |
| City | 0.096 | 0.056–0.164 | 0.718 | 0.471–1,096 |
| Natural | 0.036 | 0.024–0.051 | 0.124 | 0.082–0.187 |
| Rural | 0.004 | 0.001–0.012 | 0.027 | 0.011–0.064 |

Note:
Estimated marginal means obtained from generalized linear models (see Table 2) for the relative abundance (number of individuals/km) of each parrot species in different habitats.

best-preserved areas in the country, where we only counted 137 Hispaniolan parakeets, 15 Hispaniolan amazons and seven Jamaican parakeets gathering together at sunset (Fig. 2).

## Conservation threats

Regarding the perception of local people of the threats and conservation status of parrots, people living in 12 distant villages located in natural areas said that parrots were abundant in the past but that they are currently extinct in their area due to overharvesting. They indicated that it is now necessary to travel to the most inaccessible sites within protected areas in order to view or poach parrots. Three people also living in distant natural areas indicated that hunting as a food source was the main cause of decline of parrots. They described how parakeets and amazons are considered game species (despite prohibition of this activity) and that large flocks were often hunted for food coinciding with the shooting of massive numbers of several species of pigeons. Moreover, one farmer explained how both parakeets and amazons are often killed to avoid crop damage, using guns and glue traps.

Regarding the illegal pet trade, we found 131 parrots in captivity (66 Hispaniolan parakeets, 63 Hispaniolan amazons, and two Jamaican parakeets; Fig. 1) thanks to the help of 51 persons living in 20 different villages and cities. Owners knew that keeping wild parrots as pets is an illegal activity, so in all cases but one pets were hidden inside their homes. Our visual examination of the plumage and growth stage of these parrots allowed us to determine that most of them (74.8%) were juvenile birds that were poached a few weeks or months prior to our visit. Attending to information provided by owners, the rest of the parrots were captured on average 4.4 years ago (range: 1–25 year). They also informed us that most parrots were poached as chicks (96.7%), often by cutting the nesting tree to gain access to the nest, while the rest were captured as adults using glue traps placed in crops. Parrot keepers also provided information on the areas where 110 parrots were poached. In most cases (95.45%), parrots were poached within six protected areas (mostly in Bahoruco and Jaragua National Parks, 59% and 14% respectively) of the Dominican Republic, while four parrots were poached in the neighboring country (Haiti). Poached parrots were sold in the villages surrounding protected areas but also were transported to distant cities for sale in local markets. Pet owners said they prefer the Hispaniolan amazon over parakeets due to their ability to imitate human speech. This fact, together with the greater scarcity of amazons in the wild (see above), makes the average price of amazons indicated by pet owners

**Table 4 Food plants of parrots.**

| Plant species | Family | A. ventralis | | P. chloropterus | | E. nana | |
|---|---|---|---|---|---|---|---|
| | | Consumed | Dispersed | Consumed | Dispersed | Consumed | Dispersed |
| Simarouba glauca | Simaroubaceae | – | – | – | – | Yes | Yes |
| Adonidia merrillii* (Vitchia merrillii) | Arecaceae | – | – | Yes | Yes | – | – |
| Dypsis (Chrysalidocarpus) lutescens* | Arecaceae | – | – | Yes | Yes | – | – |
| Prestoea acuminata | Arecaceae | – | – | Yes | Yes | – | – |
| Roystonea (hispaniolana) borinquena | Arecaceae | – | – | Yes | Yes | – | – |
| Sabal dominguensis | Arecaeae | Yes | – | Yes | Yes | – | – |
| Coccoloba uvifera | Polygonaceae | – | – | Yes | Yes | – | – |
| Inga ruiziana (Gina sp.) | Leguminosae | – | – | Yes | Yes | – | – |
| Pithecellobium dulce | Leguminosae | Yes | – | – | – | – | – |
| Tamarindus indica* | Leguminosae | – | – | Yes | Yes | – | – |
| Mangifera indica* | Anacardiaceae | Yes | – | Yes | Yes | – | – |
| Sapindus saponaria | Sapindaceae | – | – | Yes | Yes | – | – |
| Melicoccus bijugatus | Sapindaceae | – | – | Yes | | – | – |
| Terminalia catappa | Combretaceae | – | – | Yes | Yes | – | – |
| Sideroxylon foetidissimum | Sapotaceae | Yes | – | – | – | – | – |
| Azadirachta indica | Meliaceae | Yes | Yes | Yes | | | |
| Casuarina sp.* | Casuarinaceae | Yes | – | – | – | – | – |
| Araucaria heterophylla* | Araucariaceae | – | – | Yes | – | – | – |
| Unknown | Unknown | Yes | Yes | – | – | – | – |

Note:
List of plant species and families of which we observed the three species of parrots consuming and dispersing fruits. Asterisks indicate non-native plant species.

(63.5 $, range: 10.7–139 $, $n = 45$) twice as high as that of parakeets (31 $, range: 10.7–64 USD, $n = 15$), to the point that in two cases poachers sold very young parakeets as amazons.

## Seed dispersal

We observed the three species of parrots feeding on a total of 19 plant species from 11 families of trees and palms, of which 14 species (74%) were dispersed by parrots by transporting fruits in their beaks to distant perching sites (Table 4). Five of the dispersed plant species were non-native to Hispaniola Island (Table 4). Given the scarcity of parrots in rural and natural habitats, all but one of the dispersal events was recorded in cities. We did not observe other species capable of dispersing these fruits through stomatochory in the urban areas surveyed. The structure of forested urban areas, urban parks and gardens mixed among buildings made it difficult to measure seed dispersal frequencies and exact dispersal distances (e.g., we could observe an amazon carrying a fruit in the beak but could not determine the exact tree from which the fruit was picked). We thus looked for trees in which both amazons and parakeets perched to handle and consume the fruits, and obtained minimum dispersal distances as the distance from each discarded seed to the nearest fruiting plant of the same species. In this way, we

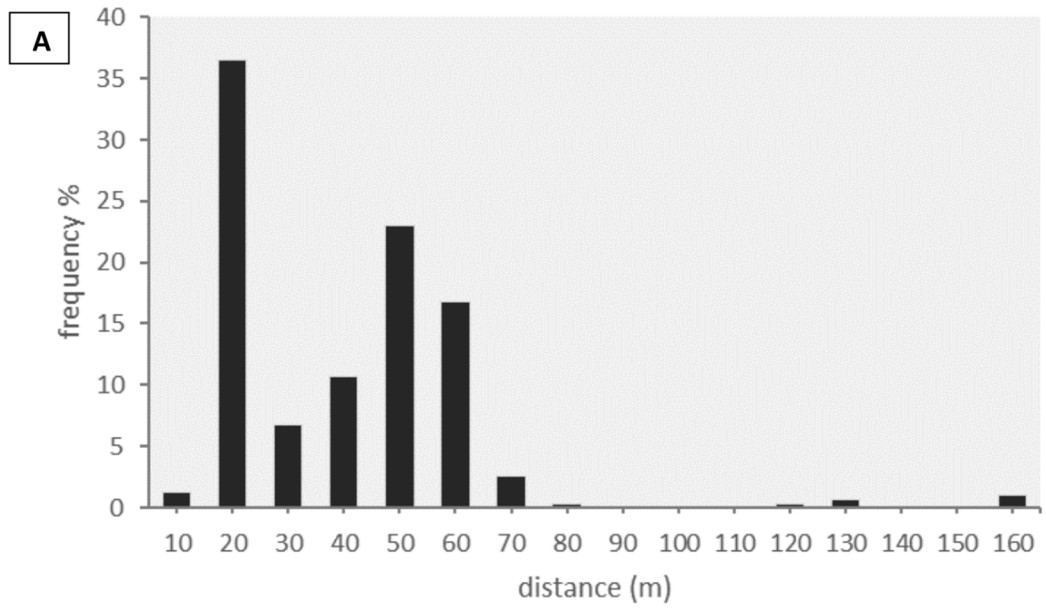

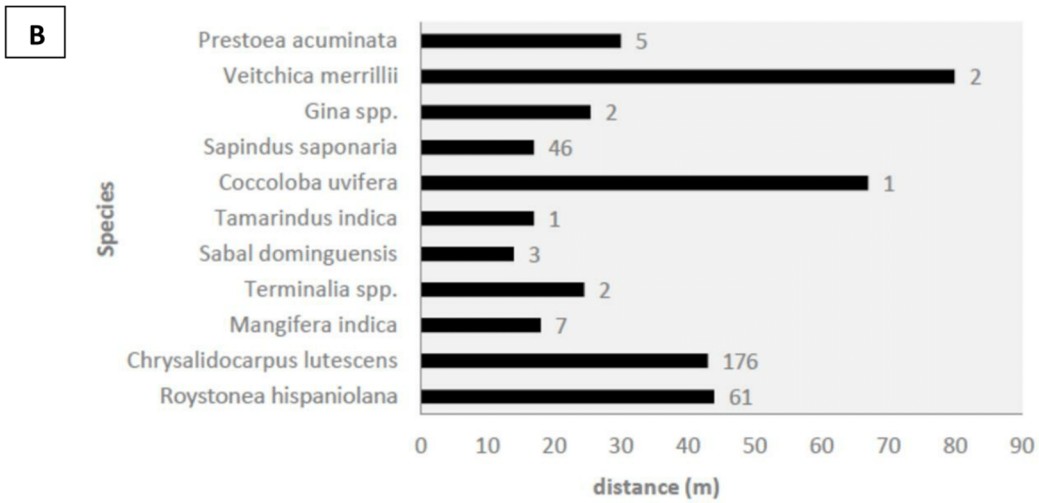

**Figure 3 Seed dispersal.** (A) Proportion of seeds dispersed by parrots ($n = 306$) grouped into ten-meter distance intervals. (B) Median dispersal distances for each plant species, with sample sizes indicated to the right of the bars.

recorded 306 dispersed seeds under 66 perching sites, corresponding to 11 plant species (Fig. 3). In 99.5% of the cases, undamaged mature seeds were discarded after pulp consumption by parrots, thus maintaining the potential for germination, with the rest being damaged or unripe seeds. The median minimum dispersal distance was 37 m (range: 8–155 m, Fig. 3A). Most of the seeds (93.85%) were dispersed to minimum distances ranging between 20 and 60 m, while only a small fraction (4.85%) was

dispersed >60 m (Fig. 3A). Minimum dispersal distances varied among plant species (Fig. 3B), but differences were not statistically significant (Kruskall–Wallis test, $\chi^2 = 9.95$, d$f = 9$, $p = 0.35$).

## DISCUSSION

### Large-scale parrot population declines

Our large-scale field survey, conducted at the end of the breeding season (*Latta et al., 2006*) when parrot population sizes are expected to be the largest due to the recent recruitment of fledglings, shows an extremely low abundance of the two parrot species endemic to Hispaniolan Island in their natural habitats. Their scarcity in the wild is striking when compared to the abundances of congeneric species we obtained following the same methodology in other Neotropical habitats. The relative abundance of Hispaniolan parakeets in natural habitats (Table 3) resulted one order of magnitude lower than that of blue-crowned parakeets (*Tecthocercus* = (*Psittacara*) *acuticaudatus*, with an abundance ×47 times higher) and mitred parakeets (*P. mitratus*, ×25.8 times higher) in Bolivian Andean dry forests (*Blanco et al., 2015*), and white-eyed parakeets (*P. leucophthalmus*, ×57.8 times higher) in Brazilian Atlantic forests (*Tella et al., 2016b*). On the other hand, the relative abundance in natural habitats of the Hispaniolan amazon resulted one to two orders of magnitude lower than that of two amazon species also listed by IUCN as Vulnerable, the red-spectacled amazon (*A. pretrei*, ×109 times higher) and the vinaceus amazon (*A. vinacea*, ×23 times higher), in the Atlantic Brazilian forests (*Tella et al., 2016b*). Even the blue-fronted amazon (*A. aestiva*) living in the inter-Andean valleys of Bolivia, where populations are suffering from strong poaching pressure for the domestic pet trade (*Pires, Herrera & Tella, 2016*), showed a relative abundance 3.3 times higher (*Blanco et al., 2015*) than the Hispaniolan amazon.

The current low abundance of Hispaniolan parrots cannot be attributed to island versus continental environmental conditions, as densities of bird species are often higher on islands than on the mainland (*Newton, 2003*), but rather to a long-term process of human-induced population decline. Hispaniolan amazons and parakeets were known to form flocks of hundreds and thousands, respectively, until 1930, with the recent decline attributed to habitat loss, hunting and trapping for the pet trade (*Latta et al., 2006*). It is worth noting that the low abundances we obtained in natural habitats cannot be explained by habitat loss, since they were mostly obtained within protected and remote areas where the habitat still is reasonably well conserved. On the other hand, hunting and trapping have been at play for centuries. The Amerindians already trapped parrots for household pets and hunted them as they were highly appreciated as a food source, and Spanish colonists also hunted and traded parrots to the point of causing the extinction of one species (*Wiley & Kirwan, 2013*).

As a matter of concern, hunting and trapping are still threatening parrot populations. Both species of parrots formed large flocks in the past that probably moved through the island while tracking food resources (*Latta et al., 2006*). According to our conversations with local people, they were often killed for food during the post-breeding season coinciding with the massive hunting of very large flocks of pigeons, which gathered

in coniferous forests during pine fructification. The access to modern guns probably accelerated the decimation of both parrots and pigeons; in fact, we only observed seven of the 10 pigeon species occurring in Hispaniola Island, and always in very low numbers. Moreover, the loss of natural habitats by agriculture may have increased crop damage by parrots and thus motivated the shooting and trapping of parrots in agricultural lands. This may explain their lowest abundance in rural habitats, despite the high food availability in the form of cereal crops and fruiting trees surrounding small villages.

Parrots are long-lived species with slow reproduction rates (*Young et al., 2012*), and thus the overharvesting of adults (through hunting and trapping) has a higher impact on population dynamics than nest poaching (*Pires, Herrera & Tella, 2016*; *Valle et al., 2018*). Nest poaching to supply the pet trade has ancestral cultural roots in Hispaniola Island (*Wiley & Kirwan, 2013*; *White et al., 2011*). However, the human population and economic growth in recent decades may have increased its intensity and contributed to the decimation of wild parrot populations. Information provided by local people supports the results of our field survey: they indicated local extinctions and population reductions in natural areas, with the last wild populations currently restricted to the protected and more inaccessible areas to which poachers are now forced to travel to obtain chicks for illegal trade. The fact that the pet trade is still a thriving activity, despite the rarity of parrots in the wild and its prohibition since 2000, may be explained both by cultural influences maintaining a high demand for pets (*White et al., 2011*) and by the lucrative benefits. Poachers may take risks since just two nests of amazons can yield monetary rewards equal to the average wage per month in the country (259 $, http://www.salaryexplorer.com/salary-survey.php?loc=61&loctype=1#disabled).

Our baseline survey suggests that the conservation status of the endemic Hispaniolan parrots is worse than previously thought (*Latta et al., 2006*; *BirdLife International, 2016a*, *2016b*), and that these species could enter a vortex of extinction in the wild if current threats are not halted. We hope this work will encourage further, more detailed conservation-aimed research on the remaining wild populations, as well as the enforcement of current laws against hunting and parrot poaching and the creation of education and conservation programs, ideally involving local residents (*White et al., 2011*).

## Cities as conservation hotspots

The scarcity of Hispaniolan parrots in natural habitats contrasts with their relatively high abundances in the two larger cities. Parrots are known to be good natural colonizers of urban habitats (*Carrete & Tella, 2011*), and the presence of urban parrots in the largest city (Santo Domingo) is known for at least three decades (C. Cano, 2017, personal communication). We found that cities offer sufficient resources, such as food (including a variety of native and exotic plants), nesting sites (cavities in trees and historic buildings, Fig. 1B) and safe roosting sites, allowing parrots to develop their complete annual cycles within urban areas. After initial urban colonization, ecological conditions differing between natural and urban habitats should explain the largest abundance of parrots in cities. Predation release (i.e., the generally lower abundance of avian predators in urban habitats, *Díaz et al., 2013*) has been shown to explain the positive population

growth of a bird species in the city up to much higher densities than in the surrounding natural habitats (*Rebolo-Ifrán, Tella & Carrete, 2017*). In the case of Hispaniolan parrots, the few avian predator species are scarce and mostly prey on small birds (*Latta et al., 2006*), and thus humans can be considered as the main "predators" of parrots through hunting and trapping of adults and nest poaching. These illegal activities are undoubtedly hampered in populated cities, where moreover people may be more prone to conserving parrots. The same predator-release mechanism could explain the proliferation in cities of an introduced exotic pigeon (*Luna et al., 2018*), in contrast to the poor conservation status of native pigeons due to hunting (*Latta et al., 2006*). In addition to natural colonization and intrinsic growth of predator-free urban populations, the release of parrots in cities may be reinforcing them; some pets could escape from cages, and 28 Hispaniolan parrots and 132 Hispaniolan parakeets seized by the police have been released in Santo Domingo since 2011 after being recovered in the city's zoo (M. Sánchez, 2017, personal communication).

Whatever the mechanisms for explaining urban parrot populations, large cities now constitute key conservation hotspots for these two endemic, globally threatened parrots. Particularly, Santo Domingo could hold the largest world population of Hispaniolan parakeets, which we estimated at c. 1,600 individuals after censusing the only known communal roost that most likely concentrates all parakeets living in the city. As recently claimed for several threatened Australian species (*Ives et al., 2016*), national conservation policies should integrate urban populations when planning for and managing these threatened parrots. Conservationists and policymakers need to understand the opposing trends of natural and urban populations, since the flourishing urban populations may mask the poor conservation status of the species in the wild.

## Conserving species and their ecological functions

The importance of not only saving species but also their ecological functions from extinction is gaining increasing support (*Valiente-Banuet et al., 2015*), to the point that some have claimed that rewilding management actions should include the introduction of non-native species that are functionally similar to extinct ones (*Corlett, 2016*). The conservation value of some non-native urban populations of threatened parrots (*Gibson & Yong, 2017*; *Mori et al., 2017*) has been recently highlighted, as they can be viewed as genetic and population stocks for conservation programs rather than just as invasive species (*Gibson & Yong, 2017*). This is a valuable conservation argument but does not take into account that ecological functions of parrots may be lost in nature if they are confined to cities, both within and outside of their native ranges. This is likely because their ecological functions, other than their role as plant antagonists, have been largely overlooked until recently (*Blanco, Hiraldo & Tella, 2018*).

Despite the short-term nature of our baseline survey, we were able to demonstrate that Hispaniolan parrots are legitimate long-distance seed dispersers of most of their food plant species, as they frequently transport fruits to distant perching trees where they discard undamaged seeds after fruit consumption. These results should be taken with caution, since most seed dispersal records were obtained from food plants growing in

urban habitats. Unfortunately, to our knowledge there are no studies focusing on the diet and foraging behavior of these species in the wild. However, it is reasonable to expect Hispaniolan parrots are also acting as frequent seed dispersers in natural habitats, as we have recently shown for other parrot-plant systems (*Blanco et al., 2015*; *Blanco, Hiraldo & Tella, 2018*; *Tella et al., 2015*, *2016a*, *2016b*; *Baños-Villalba et al., 2017*; *Montesinos-Navarro et al., 2017*). Notably, parrots are also covering an important ecological function in urban environments, but with the undesirable side effect of often spreading exotic plant species (see Table 4). On the other hand, it is worth noting that we only focused on the easily observed external dispersal (stomatochory), while internal dispersal through the ingestion and defecation of small viable seeds (endozoochory) is also expected to occur as in other parrot species (*Blanco et al., 2016*). Other mutualistic functions of Hispaniolan parrots can be also expected, such as pollination, food wastage that facilitates secondary dispersal and food for terrestrial animals, and the consumption of plant parasites (*Montesinos-Navarro et al., 2017*; *Blanco, Hiraldo & Tella, 2018*).

Parrots have been shown to be key dispersers of some plant species in contrasting ecosystems (*Boehning-Gaese, Gaese & Rabemanantsoa, 1999*; *Blanco et al., 2015*; *Tella et al., 2016a*, *2016b*; *Baños-Villalba et al., 2017*), and their regional or global extinction may disrupt ecological processes with uncertain consequences (*Blanco, Hiraldo & Tella, 2018*). One of the ecological functions most affected by the decline of plant-animal mutualisms is seed dispersal (*Howe & Smallwood, 1982*), in the way that the disruption of disperser-plant interactions can trigger declines in plant diversity (*Cordeiro & Howe, 2003*), seedling recruitment (*Terborgh et al., 2008*), and gene flow in fragmented landscapes (*González-Varo et al., 2017*). Disruptions of seed dispersal by habitat loss and defaunation can be more dramatic in the case of islands (*Traveset, Gonzalez-Varo & Valido, 2012*; *Fontúrbel, Jordano & Medel, 2017*), given a generally lower species diversity than in the mainland (*Newton, 2003*) and thus a higher probability of losing functionally non-redundant species (*Valiente-Banuet et al., 2015*). In the case of Hispaniola Island, the only large-sized species feeding on fruits are one trogon and two crow species, which are scarce and threatened, while the rest are small passerines (*Latta et al., 2006*). These smaller-sized species are expected to only disperse tiny seeds through endozoochory, and thus parrots are probably the only species capable of dispersing the seeds of plants producing large fruits and seeds through stomatochory (see *Blanco et al., 2016*). Therefore, the loss of rare species performing rare functions, as may be the case of Hispaniolan parrots, may have a stronger impact on ecosystem functioning (*Violle et al., 2017*). Unfortunately, the regional and global extinction of these parrot species are not a mere possibility, but a likely fact. Parrot populations began a strong decline in the Caribbean islands upon European colonization due to habitat loss, hunting and the pet trade (*Wiley, 1991*), to the point that endemic macaws inhabiting up to 11 islands were all extinct by the 1850s (*Wiley & Kirwan, 2013*), as well as subspecies of the Hispaniolan parakeet endemic to Puerto Rico (by 1900), the Guadeloupe amazon (*A. violacea*) and the Martinique amazon (*A. martinicana*) were extinct as a result of hunting (by the end of the 18th century), while 10 out of the 13 extant Caribbean parrot species are globally threatened (http://www.iucnredlist.org, *IUCN, 2017*).

## CONCLUSIONS

If current trends continue, approximately five billion of the world's eight billion residents will live in cities by 2030, with projections of nearly six million square kilometers of land converted to urban areas (*Seto, Güneralp & Hutyra, 2012*). This global urbanization process will increase the negative impacts on biodiversity through habitat loss (*Seto, Güneralp & Hutyra, 2012*; *Newbold et al., 2015*), but also the role of cities as conservation hotspots for a number of threatened species that perform better there than in their natural habitats (*Rebolo-Ifrán, Tella & Carrete, 2017*). Our example using Hispaniolan parrots as a case study may reflect many others, currently overlooked or expected in the near future, given the widespread human impact on parrot populations through habitat loss and overharvesting (*Tella & Hiraldo, 2014*; *Olah et al., 2016*; *Berkunsky et al., 2017*). On the other hand, the role of Hispaniolan parrots as seed dispersers should not come as a surprise, given the variety of ecological functions of parrots that have gone unrecognized until recently (*Blanco, Hiraldo & Tella, 2018*). Therefore, although conservation planning should seriously consider the value of urban populations of parrots and many other threatened taxa (*Ives et al., 2016*; *Gibson & Yong, 2017*), it should not distract efforts to restore their populations in natural habitats to conserve their ecological functions. In fact, the loss of ecosystem functions and services through local biodiversity loss should be given as much attention as the rate of species extinction in the current scenario of global change (*Newbold et al., 2012*).

## ACKNOWLEDGEMENTS

Thanks to Cyntia Ortiz of the Ministry for her help during the preliminary stages of the expedition, the people from Asociación Jaragua for their help in the field, the National Botanical Garden of Dominican Republic for the identification of plants, and the National Zoological Park for their advice and information. We are also grateful to David Aragonés (LAST-EBD) for helping us to elaborate the map. Two anonymous reviewers greatly helped to improve our manuscript.

### Funding

This work was funded by project CGL2015-71378-P from MINECO (Spain). Álvaro Luna was supported by La Caixa-Severo Ochoa International PhD Program 2015 and the Severo Ochoa Program for Centres of Excellence in R+D+I (SEV-2012-0262). The funders had no role in study design, data collection and analysis, decision to publish, or preparation of the manuscript.

### Grant Disclosures

The following grant information was disclosed by the authors:
Project CGL2015-71378-P from MINECO.
La Caixa-Severo Ochoa International PhD Program 2015.
Severo Ochoa Program for Centres of Excellence in R+D+I (SEV-2012-0262).

## Competing Interests

The authors declare that they have no competing interests.

## Author Contributions

- Álvaro Luna conceived and designed the experiments, performed the experiments, analyzed the data, prepared figures and/or tables, authored or reviewed drafts of the paper, approved the final draft.
- Pedro Romero-Vidal conceived and designed the experiments, performed the experiments, analyzed the data, prepared figures and/or tables, authored or reviewed drafts of the paper, approved the final draft.
- Fernando Hiraldo conceived and designed the experiments, performed the experiments, contributed reagents/materials/analysis tools, authored or reviewed drafts of the paper, approved the final draft.
- Jose L. Tella conceived and designed the experiments, performed the experiments, contributed reagents/materials/analysis tools, authored or reviewed drafts of the paper, approved the final draft.

## Field Study Permissions

The following information was supplied relating to field study approvals (i.e., approving body and any reference numbers):

Field surveys were approved by Ministerio de Medio Ambiente y Recursos Naturales de la República Dominicana.

## Data Availability

The raw data are provided in the Supplemental Files.

## Supplemental Information

Supplemental information for this article can be found online at http://dx.doi.org/10.7717/peerj.4908#supplemental-information.

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
