# Peer review of "Cities may save some threatened species but not their ecological functions"

_PeerJ, doi:10.7717/peerj.4908_

## Round 0.1 · original submission · Major Revisions

Please address each comment and suggested edit by the two reviewers either by incorporating them or clearly explaining why they were not incorporated.

Reviewer 1 ·

Basic reporting

This paper is clearly written and is a pleasure to read. While much of the relevant literature has been cited, I think the authors could provide more background to justify their methods and convince the reader of the robustness of their data.

Figure 2: On the map, would it be possible to indicate the different habitat types?

Experimental design

The research questions are well defined, and are certainly meaningful in the context of the conservation of these two parrot species. The paper addresses an interesting point that despite the conservation value of urban populations for the survival of threatened species, urban populations cannot protect against the loss of important ecological functions in the native habitats. While in general I find the story convincing, I find much of the data to be somewhat anecdotal as presented. The authors could do more to justify their choice of methods and to convince the reader of the validity of their findings by providing more detail of their methods, backed up with relevant literature.

It would be helpful to explain how the ad-hoc observational data of parrot/parakeet presence were included in the statistical analysis. The data from the roadside surveys clearly reflect that urban populations of the parakeet are larger than rural populations, but for the Amazons there is a relatively even distribution between the city and two tropical forest types. I am a bit surprised that the GLMM would have showed a significant effect of habitat given this data as presented in Table 1. I can see that the numbers skew towards higher populations in urban areas if the off-survey observational counts are included in the analysis, so it would be helpful to clarify if this data were or were not a part of this result.

Validity of the findings

While I understand at this point that it would be difficult to go back and collect more data, I do think that a more thorough reporting of the literature on the various methods used could help the reader to assess the strength of the data provided. For example, how confident are the authors that the numbers recorded on the roadside surveys represent an accurate estimate of actual population numbers? Could it be that urban parrots flock more and so higher numbers are observed? (This point could likely be addressed by reporting on the average number of individuals observed in a single observation event.) I imagine that parrots would be more easily detected in urban areas, can the data provided somehow account for that detectability? Perhaps including information on total habitat patch area would help to address this? Were all transect routes only surveyed once, or did the authors do repeat surveys to account for differences in detectability on different days? This seems like it would be particularly important for the forested habitats given that the chance of detecting parrots at a single site in a large forest are surely affected by chance in some ways. There may be some good methods for accounting for this which the authors should flesh out more fully in the methods.

The interview data provides an interesting context for this story and in general I believe the findings are true. But while I fully understand the difficulty of collecting data on illegal activities, more could be done to increase the validity of the data collected. At the very least, more information should be provided on the interview methods used, including types of questions asked and any way that they judged the validity of the answers being provided to them. Also, it is common to conduct pilot interviews in order to improve the interview design and perhaps the authors could comment on whether any pilot studies were conducted, or whether their interview technique changed over the course of the study. I want to emphasize here that I am not criticizing the use of the interview technique nor do I have any strong reason to doubt the data. I am primarily pointing out that there are many methods available to improve the validity of this type of data which the authors might consider. I would refer the authors to a recent paper which addresses some of these concerns and suggests methods for increasing confidence in this type of data: Young et al. 2018. A methodological guide to using and reporting on interviews in conservation science research. Methods in Ecology and Evolution 9:10-19.

Seed dispersal was estimated in part by searching below trees in which parrots were perched to identify dispersed fruits and seeds. Please elaborate on how you were able to determine which seeds were dispersed by the parrots rather than consumed and dropped from that tree, or dropped by other animals/naturally. Did you observe seeds being dropped by parrots? If not, how could you differentiate between a seed dropped by a parrot, or consumed by another species once the fruit had fallen to the ground? Also regarding the seed dispersal, is there any data at all on what they eat in the wild? Are there native species in the forest habitats where the parrots occur that they would be treating in the same way as the urban species? I believe that some data on foraging in natural areas was collected, it would be good to report that here even if the sample sizes are low.

The discussion is very long and could be shortened considerably.

Additional comments

Overall I enjoyed reading this paper which is well written and contains an interesting set of data. I think the paper addresses an interesting and valid concern about the focus on the conservation value of urban populations by cautioning us not to forget about the loss of ecological function if these species no longer occur in their natural populations. While in general I find the story convincing based on the evidence provided, I find that much of the data are fairly anecdotal and more could be done to increase the robustness of the results. At least there could be more discussion to address the potential weaknesses in the data provided. Likewise, I would suggest tempering some of the stated conclusions based on the strength of the data.

Line 98 – replace “assumptions” with “hypotheses”

Line 119 and 327 – Find a more suitable reference than Wikipedia!

Line 225 – change previous to “prior”

Line 245 – not clear how pigeons come into this, more explanation here would be useful.

Reviewer 2 ·

Basic reporting

This is a very interesting study with broad implications for the conservation of biodiversity within cities. The authors provide an excellent introduction to the focus of the paper and the biology of the species of interest. The writing is generally very well done, with nice cohesive paragraphs and excellent use of references. I do identify some shortcomings with respect to the scientific foundation for the study. Though, I believe these can be remedied with revision.
I have included some suggestions for added referencing below.
line 55-57: awkward wording - suggest rephrasing.
line 62-64: indeed, there will be a shift in functioning and services, but could be the case that they'll offer new services or disservices in urban environments. See: Potgieter, L. J., Gaertner, M., Kueffer, C., Larson, B. M., Livingstone, S. W., O’Farrell, P. J., & Richardson, D. M. (2017). Alien plants as mediators of ecosystem services and disservices in urban systems: a global review. Biological Invasions, 19(12), 3571-3588.
Granted, this is for alien plant species, but many of the same principles apply for a shift in the distribution of native and/or threatened species to urban areas.
line 74-78: This is really interesting. On a philosophical tangent: Do we think about our pets as providing ecosystem services (e.g. cultural appreciation)? or do they cease to provide those services once they've been domesticated?

My main criticism of the study is in the lack of hypotheses that would otherwise guide analysis and discussion. The final paragraph of the introduction is the place to do this, but the authors, oddly, instead choose to highlight findings of the study. This final paragraph needs to be re-written to include hypotheses related to each line of inquiry in the analysis. This is where you should clearly state the hypotheses guiding the study and the general methodological approach, with more detailed methods following in the next section. I do believe this is possible to do with a revision.

Experimental design

- would recommend not using wikipedia as a reference.
line 131: "habitat use by parrots" - these "metrics" (also relative abundance in relation to habtiat, conservation threats, seed dispersal) should be introduced, with hypotheses, in the final paragraph of your introduction, then defined in the methods section.

For the habitat assessment, it is unclear how all this data is being synthesized. Perhaps a couple sentences on what you did with this data.

line 154-165: Would you expect parrot density to be significantly affected (positively or negatively) by proximity to roads? Probably need to comment on that.

line 167-180: This method/section would do well to reference some social-scientific approaches (really should have been used at the onset to develop your methods, but at least you can cite the scientific approaches that should have been used. (i.e. stakeholder analysis). See:
Reed, M. S., Graves, A., Dandy, N., Posthumus, H., Hubacek, K., Morris, J., ... & Stringer, L. C. (2009). Who's in and why? A typology of stakeholder analysis methods for natural resource management. Journal of environmental management, 90(5), 1933-1949.
&
Prell, C., Hubacek, K., & Reed, M. (2009). Stakeholder analysis and social network analysis in natural resource management. Society and Natural Resources, 22(6), 501-518.

Validity of the findings

The authors present some compelling findings regarding the distribution and density of these parrots in different habitats throughout the region. Again, these findings have significant implications for the future of biodiversity in cities, in general. The discussion is very well developed, with many interesting points regarding ecological function, the role of rare species and the potential for non-native species to take on the role of endemic species that are in decline.

Again, my main criticism is that the study needs to driven by scientifically-grounded hypotheses, and that these findings should be discussed in relation to those hypotheses.

Regarding the social aspects of the study, hypotheses may be difficult, but certainly a more formal framework/method for analysis is needed.

- line 218: Is this data including those parrots identified in homes? please clarify. If so, this definitely stands to affect much of your discussion.

-line 221: How did you decide which homes to sample? Was there a systematic approach? unclear.

-line 238: This line of inquiry should have been introduced earlier.

-table 1: check format

---

## Round 0.2 · accepted · Accept

Thank you for clearly addressing comments made previously by the reviewers.

# Reviewer 1 ·

Basic reporting

After revisions, this paper, and in particular the hypotheses, are much more clear. Overall the paper is well written with sufficient background provided and reference to the relevant literature.

Experimental design

In this version of the manuscript, the methods are explained more clearly and the choice of methods is better justified.

Validity of the findings

Data is robust and statistically sound. Conclusions are well stated (and not over-stated).

Additional comments

I am happy with the revisions in this version of the manuscript and am pleased to give my support to the publication of this paper. I agree that it contains information important to the conservation of parrots.